# Identifying MicroRNAs Suitable for Detection of Breast Cancer: A Systematic Review of Discovery Phases Studies on MicroRNA Expression Profiles

**DOI:** 10.3390/ijms242015114

**Published:** 2023-10-12

**Authors:** Lisa Padroni, Laura De Marco, Valentina Fiano, Lorenzo Milani, Giorgia Marmiroli, Maria Teresa Giraudo, Alessandra Macciotta, Fulvio Ricceri, Carlotta Sacerdote

**Affiliations:** 1Unit of Cancer Epidemiology, Città Della Salute e Della Scienza University-Hospital and Center for Cancer Prevention (CPO), Via Santena 7, 10126 Turin, Italy; lisa.padroni@edu.unito.it (L.P.); ldemarco@cittadellasalute.to.it (L.D.M.); giorgia.marmiroli@edu.unito.it (G.M.); 2Unit of Cancer Epidemiology, Department of Medical Sciences, University of Turin, 10126 Turin, Italy; valentina.fiano@unito.it; 3Centre for Biostatistics, Epidemiology and Public Health (C-BEPH), Department of Clinical and Biological Sciences, University of Turin, 10043 Orbassano, Italy; lorenzo.milani@unito.it (L.M.); mariateresa.giraudo@unito.it (M.T.G.); alessandra.macciotta@unito.it (A.M.); fulvio.ricceri@unito.it (F.R.)

**Keywords:** breast cancer, microRNA, miRNA, serum, plasma, high throughput techniques

## Abstract

The analysis of circulating tumor cells and tumor-derived materials, such as circulating tumor DNA, circulating miRNAs (cfmiRNAs), and extracellular vehicles provides crucial information in cancer research. CfmiRNAs, a group of short noncoding regulatory RNAs, have gained attention as diagnostic and prognostic biomarkers. This review focuses on the discovery phases of cfmiRNA studies in breast cancer patients, aiming to identify altered cfmiRNA levels compared to healthy controls. A systematic literature search was conducted, resulting in 16 eligible publications. The studies included a total of 585 breast cancer cases and 496 healthy controls, with diverse sample types and different cfmiRNA assay panels. Several cfmiRNAs, including MIR16, MIR191, MIR484, MIR106a, and MIR193b, showed differential expressions between breast cancer cases and healthy controls. However, the studies had a high risk of bias and lacked standardized protocols. The findings highlight the need for robust study designs, standardized procedures, and larger sample sizes in discovery phase studies. Furthermore, the identified cfmiRNAs can serve as potential candidates for further validation studies in different populations. Improving the design and implementation of cfmiRNA research in liquid biopsies may enhance their clinical diagnostic utility in breast cancer patients.

## 1. Introduction

MicroRNAs (miRNAs) constitute a class of small RNA molecules that are naturally present and have been conserved over evolutionary history [1]. These single-stranded RNA molecules do not participate in the encoding of proteins and typically consist of 19 to 25 nucleotides [1]. A collection of around 2650 distinct mature microRNA sequences is documented in miRNA libraries [1]. Functionally, miRNAs play a critical role as post-transcriptional regulators, influencing gene expression across various tissues and developmental stages. They accomplish this by engaging in precise interactions within intricate regulatory networks [2].

Due to their limited binding region between miRNA and mRNA, a single miRNA has the capacity to target multiple specific mRNAs, thereby exerting influence across diverse pathways [2]. Given their diverse functions, miRNAs possess the ability to regulate various pathways associated with cellular activities and intercellular communication. These pathways encompass processes such as cellular growth, specialization, replication, and programmed cell death [3].

Approximately half of the genetic codes responsible for miRNAs in humans are situated within regions of the genome that are linked to cancer or at chromosomal sites prone to fragility and instability [4].

In breast cancer, as in numerous other cancer types, the onset of abnormal cell behavior leads to uncontrolled proliferation. This proliferation is driven by genetic modifications that influence cellular growth regulatory mechanisms. The miRNAs associated with this disease can be classified into two categories: oncogenic miRNAs (known as oncomiRs) and tumor suppressor miRNAs (referred to as tsmiRs) [1]. OncomiRs are generally upregulated in breast cancer and function by suppressing the expression of potential tumor-suppressing genes [5]. Conversely, tsmiRs hinder the expression of oncogenes that contribute to the formation of breast tumors [5]. Consequently, decreased expression of tsmiRs can lead to the initiation of breast malignancy [5]. Figure 1 offers an overview of the specific regulatory roles of miRNAs in breast cancer.

These regulatory networks encompass several fundamental aspects of cancer biology, including the maintenance of growth signals that promote proliferation, the achievement of replicative immortality, the initiation of invasion and metastasis, the resistance to programmed cell death and apoptotic responses, the stimulation of new blood vessel formation (angiogenesis), the activation of cellular metabolism and energy processes, and the facilitation of immune evasion by cancer cells [5]

Liquid biopsy provides important information on the analysis of circulating tumor cells and circulating tumor-derived materials, such as circulating tumor DNA, circulating miRNAs (cfmiRNAs), and extracellular vesicles [6].

In particular, cfmiRNAs have been extensively investigated as diagnostic biomarkers, other than as biomarkers for prognosis and therapy response. CfmiRNAs constitute a group of short, noncoding regulatory RNAs that modulate gene expression at the post transcriptional level [7]. Cell-free circulating microRNAs likely released from cells in lipid vesicles, microvesicles, or exosomes have been detected in peripheral blood circulation [8].

Usually, the study design of research works on biomarkers consists of a first phase generally regarded as a discovery phase, followed by a validation phase [9].

The discovery phase typically involves exploration carried out with high-throughput laboratory techniques to select a pool of candidates [10]. The objective is to identify a short list of promising cfmiRNAs associated with disease for further investigation. The discovery research poses considerable challenges, due to the large number of biomarkers being investigated, the typical weakness of signals from individual markers, and the frequent presence of strong noise due to experimental effects [10]. The validation study is a key step for translating laboratory findings into clinical practice; furthermore, this is heavily conditioned by the short list of biomarkers selected in the discovery phase [10].

While evolving molecular technologies in discovery studies have generated plenty of omics data, identification success has been very limited considering the reduced number of cfmiRNAs that have reached clinical use [11].

One of the reasons behind this phenomenon is the lack of adequate study designs in the discovery phase research [12]. Furthermore, several studies analyze candidate cfmiRNAs selected from a search on previous literature, thereby amplifying the problems that may have arisen due to a suboptimal discovery phase.

The search for cfmiRNAs to use as diagnostic biomarkers in breast cancer is very active. Several reviews and meta-analyses have been published on the predictive role of cfmiRNAs in breast cancer diagnosis [13,14,15,16,17]. Nevertheless, all of them were based on validation phases of the study or on studies on candidate cfmiRNAs.

This review aims to identify the altered levels of circulating microRNA in breast cancer patients compared to healthy controls, including only the discovery phases of the study. This can be of great usefulness for the progression of this research field, allowing the selection of candidate cfmiRNAs to be investigated in new case–control studies.

## 2. Materials and Methods

We have registered the protocol of this review in the international database of prospective registered systematic reviews (PROSPERO 2022; CRD42023399977). The workflow and methodology were based on the Preferred Reporting Items for Systematic Review and Meta-Analyses of Diagnostic Test Accuracy (PRISMA-DTA) guidelines [18].

### 2.1. Publication Search

We capitalized on a previous literature review conducted by our group, in which we conducted searches on PubMed, Cochrane Library, EMBASE, Google Scholar, and NCBI PubMed Central to select appropriate studies [13] (the previous review was updated to 31 December 2022).

The search was performed using the following keywords as a search strategy: ((Circulating) AND (microRNA OR miRNA) AND (breast AND Cancer)) NOT (cells) NOT (tissue) AND ((English [Filter]) AND (Humans [Filter]) AND (“31 December 2022” [Date—Release])). Additionally, other studies were identified through the references in previously selected publications.

### 2.2. Inclusion and Exclusion Criteria

In the systematic review, we considered all studies that fulfilled the following requirements: (1) inclusion of both patients with BC and healthy controls; (2) measurement of cfmiRNA levels in serum, plasma, or blood; and (3) presence of a discovery phase that used high throughput techniques, including studies with an agnostic genome-wide design.

Studies were excluded if they were candidate cfmiRNA studies, reviews, meta-analyses, letters, commentaries, or conference abstracts or if they were duplications of previous publications or written in languages other than English.

### 2.3. Data Extraction

Adhering to the inclusion criteria, the primary authors (L.P. and C.S.) independently gathered the relevant data. In the event of any disagreements, consensus was reached through discussion. The extracted data included first author’s name and reference, country, sample size, biological sample type (plasma, serum, or blood), cfmiRNAs, AUC value (95% CI), fold change (95% CI), and expression (upregulation or downregulation).

### 2.4. Quality Assessment

All studies included in the review underwent independent evaluation for quality by two reviewers, L.P. and C.S. They utilized the revised Quality Assessment of Diagnostic Accuracy Studies tool (QUADAS-2) [19] to assess potential biases in four critical domains: patient selection, index test, reference standard, and flow and timing. The agreement percentage between the two reviewers was calculated for each variable in QUADAS-2. Any discrepancies in coding or QUADAS-2 assessments were resolved through consensus discussions.

### 2.5. Statistical Analysis

We used STATA17.0 software to perform the statistical analyses. Pyramid plots were chosen to illustrate descriptive statistics on the directions of microRNA expression; sample subgroups were created to compare cfmiRNA expressions in different biological samples (serum and plasma).

## 3. Results

We took advantage of a previous literature review performed by our group, where from a total of 308 initially identified records, we excluded 206 records for several reasons (duplicates, secondary literature, being off topic, etc.) (see [13] for details). In total, 102 papers were considered in the screening stage for a manual review of titles and abstracts; 3 papers were excluded because the abstract was not available in English. After carefully examining the abstracts and, when useful, the full texts, an additional group of 83 publications were excluded as they did not meet the inclusion criteria (i.e., the discovery phase was performed only in tissues, or the discovery technique was not of a high-throughput type). Ultimately, this review included 16 publications [19,20,21,22,23,24,25,26,27,28,29,30,31,32,33,34,35,36]. Figure 2 illustrates the flowchart depicting the paper exclusion process.

Table 1 provides a summary of the key features of these studies. This review encompassed a total of 585 breast cancer (BC) cases and 496 healthy controls. Among the included studies, only 1 out of 16 had more than 100 BC cases [24]. The studies were conducted in various countries, including China (N = 3), the USA (N = 3), Germany (N = 2), Italy (N = 1), Ireland (N = 1), Denmark (N = 1), the Czech Republic (N = 1), Australia (N = 1), Singapore (N = 1), Malaysia (N = 1), and Saudi Arabia (N = 1). Notably, most of the studies focused on a white European population (N = 6), while the remaining studies predominantly focused on Asiatic (N = 6) or mixed U.S. or Australian populations (N = 4). This supports the evidence that Black and Hispanic populations were relatively limited in the context of microRNA and breast cancer research.

Regarding the types of samples, some studies used serum (N = 6), while others used plasma (N = 7) or whole blood (N = 3).

The 17 studies included in the review employed different panels of microRNA assays: the TLDA human micro RNA cards (N = 5) was the most popular, followed by Exiqon microRBA panel miRCURY (N = 3) and Agilent Human microarray (N = 3).

The sixth column of Table 1 presents the QUADAS domains for which a potential risk of bias was identified in each study.

The quality assessment using the QUADAS-2 tool showed that the included studies had low applicability but a high risk of bias (Figure 3). A higher risk of bias was observed in many studies across the QUADAS-2 domains of patient selection, index testing, and flow and timing (respectively, 62.5%, 100%, and 50% of studies with a risk of bias). Patient selection involves detailing the methods of selecting patients, while index testing pertains to how the cfmiRNA analysis was conducted and interpreted, standard of reference assesses the accuracy of disease status classification, and flow and timing refer to the time interval and any interventions before cfmiRNA analysis. Indeed, several studies lacked sufficient detail on the patient selection process, such as whether cases consisted of consecutive patients or controls originated from the same population that produced the cases. Furthermore, there was insufficient information on the timing of biological sample retrieval, such as whether it occurred at diagnosis, before or after surgery, or during chemotherapy. The breast cancer diagnosis was histologically confirmed in all the studies, indicating a low risk of bias in the reference standard domain. In the category of the index test, some studies failed to mention whether a threshold was pre-specified.

Furthermore, the authors of 9 out 16 studies applied a multiple testing correction in the cfmiRNA selection (mostly the Benjamini–Hochberg False Discovery Rate method), Moreover, Cuk et al. [23] and Kodhal et al. [26] also performed the adjustment for multiple comparisons, considering unadjusted *p* values for cfmiRNA selection in the validation phase.

The authors employed very heterogeneous criteria to select interesting cfmiRNAs for inclusion in the validation phase of their study. Godfrey et al. [24] and Shin et al. [32] focused on those demonstrating statistical significance in the discovery phase (*p* < 0.05). Schrauder et al. [20] selected the 25 top hits from statistically significant cfmiRNAs (*p* < 0.05). Chan et al. [22] chose cfmiRNAs with statistical significance (*p* < 0.05) excluding those with collinearity. Cuk et al. [23], Shen et al. [28], Zearo et al. [29], Zhang et al. [30], and Hamam et al. [33] used both statistical significance (all *p* < 0.05 except for Zearo *p* < 0.01) and fold change (generally FC > 2) as selection criteria. Ng et al. [25] and Jusoh et al. [34] opted for cfmiRNAs with a fold change greater than 2, while Ferracin et al. [31] selected those with the highest fold changes in plasma and serum. Wu et al. [21] focused exclusively on up-regulated cfmiRNAs (and showed them in a table) but validated only cfmiRNAs with the same pathway in serum and tissue. McDermott et al. [27] used the ANN data mining algorithm to identify cfmiRNAs with detectable and altered expression in patients. Záveský et al. [35] chose those with a Ct value exceeding 40, and finally, Kodahl et al. [26] performed automatic selection using component-wise likelihood-based boosting.

Table 2 shows the results of the studies included in this review.

To summarize the results of the studies, we decided not to discriminate between mature miRNAs originating from the opposite arms of the same precursor miRNA (i.e., we did not include suffixes such as ‘−3p’ or ‘−5p’ in the tables and figures).

The most interesting miRNAs that appear to be cfmiRNAs deserving validation in further studies are MIR16, MIR145, MIR106a, MIR193b, and MIR199a. In fact, these specific cfmiRNAs emerged in at least two independent papers for each sample type, both in serum and plasma studies as potential candidates for validation studies (Figure 4). Moreover, only MIR193b showed a coherent direction among the cases and controls.

The data in McDermott [27] were not included in Table 2 due to the lack of information on the direction, AUC, and *p*-value. Suffixes such as ‘−3p’ or ‘−5p’ are not considered in the cfmiRNA description.

The two cfmiRNAs that were selected as the most interesting in terms of coherence among studies in the previous metanalyses (MIR21 and MIR155) [13,14] emerged as statistically significant in the discovery phases only in plasma or serum, respectively.

Forty-two other cfmiRNAs other than MIR21 and MIR155 showed statistically significant different concentrations between the BC cases and healthy controls in at least two studies.

Unfortunately, the considered articles do not provide adequate data to draw a metanalysis forest plot.

## 4. Discussion

In a previous paper by our group, we conducted a systematic review of clinical studies on cfmiRNAs for the diagnosis of BC [13]. The review encompassed all studies that validated or analyzed candidate genes. In that study, we found a lack of consistency in the circulating cfmiRNAs identified across various studies. Similar results have been described in previous reviews [14,15,16,17].

This lack of replication among studies could be attributed to several factors, such as variations in the methods used for selecting cfmiRNAs, the absence of standardized techniques (including differences in sample collection and preservation, laboratory methodologies, cfmiRNA measurement and normalization, and cut-off values), inconsistent patient selection, limited cfmiRNA abundance, small sample sizes, and inadequate statistical analysis.

Recognizing the discovery phase as a potential contributor to inconsistency in the results, we performed a review of the studies that involved a discovery phases. The aim of the present work was to describe and resume the results of discovery phase studies to find the most promising cfmiRNAs that could be replicated in future candidate cfmiRNA studies. Furthermore, we will try to at least explain the lack of reproducibility of the previous candidate studies.

In general, the accurate quantification of cfmiRNAs in body fluids poses several challenges due to their low abundance and small size. This is particularly challenging for discovery studies that, in order to detect large numbers of cfmiRNAs simultaneously, use microarray profiling, quantitative RT-PCR profiling, or targeted assays of specific cfmiRNAs.

The most common biofluids used for cfmiRNA analysis are whole blood, serum, and plasma. Moreover, using the same sample type, different methods of sample preparation, anticoagulation, centrifugation, and storage properties, especially if the same high-throughput technique were used, contributed to variability and inconsistencies between reported results.

Another critical step in discovery studies is normalization, which contributes to the heterogeneity of the results. Deng et al. proposed a solution to the normalization issue which might produce more consistent results [36]; however, very few studies applied this method.

Furthermore, the same normalization issue was encountered in the collection of fold changes; they could not be compared as they were constructed using different methods, resulting in varying normalizations with the 2-delta method [37], percentage variations, and concentration ratios.

This highlights the necessity of standardized statistical analyses during discovery phases, especially when comparing cfmiRNAs concentrations between cases and controls. An illustrative example of the lack of standardization is the observed omission of multiple testing adjustment, a factor that could potentially introduce bias. In fact, implementing a *p*-value cutoff for candidate selection could introduce inflated effect sizes, thereby potentially distorting results. For this reason, it is essential to strike a careful balance between not adjusting for multiple comparison and diminishing statistical power due to the selection of a reduced number of candidate biomarkers.

Due to the considerable variability in the outcomes of cfmiRNA studies, as described above, consolidating the findings of diverse studies through systematic reviews enables an improvement in the body of evidence. In particular, five cfmiRNAs (MIR16, MIR145, MIR106a, MIR193b, and MIR199a) emerged from the discovery phases both in serum and in plasma in at least two independent papers as potential candidates for validation studies for BC diagnosis. This result is weakened by the fact that these miRNAs except one (MIR193b) showed varying counts between cases and controls, with inconsistent directions across different studies.

In addition to the necessity of conducting well-designed rigorous studies, there exists a critical need to enhance the reporting of scientific research. Checklists designed to assist authors in reporting biomarker studies, such as those provided by the STROBE-ME (Strengthening the Reporting of Observational studies in Epidemiology—Molecular Epidemiology) initiative, could significantly aid in crafting scientific papers with essential information concerning the collection, handling, and storage of biological samples; laboratory methods; the validity and reliability of biomarkers; nuances of study design; and ethical considerations [38].

Accurate and standardized reporting has the potential to greatly contribute to the accumulation of information in systematic reviews, which, in turn, can facilitate the advancement of our understanding of miRNA dynamics and their associations with various cancers.

## 5. Conclusions

The discovery phases of studies on biomarkers are crucial for identifying interesting signals to translate into clinical diagnostics. The bias encountered in this phase could cause a suboptimal discovery of new candidate biomarkers and could nullify the research effort.

For the aforementioned reason, we express our hope that forthcoming studies on cfmiRNAs that remain a promising biomarker to be implemented in liquid biopsies for BC diagnosis will have robust design and standardized procedures.

Studies including Black or Hispanic populations, other age groups, and patients with other medical conditions should be run. Additionally, it would be beneficial to capitalize on high-throughput laboratory technologies to conduct discovery studies using an appropriate sample size; to adopt a prospective design; and to adhere to standardized protocols for sample preparation, normalization, and data analysis.

Finally, researchers publishing articles on miRNAs and breast cancer should adhere to the STROBE-ME checklist when composing their papers. This approach is poised to significantly enhance the quality of their work and to propel advancements in knowledge within this domain.

## Figures and Tables

**Figure 1 ijms-24-15114-f001:**
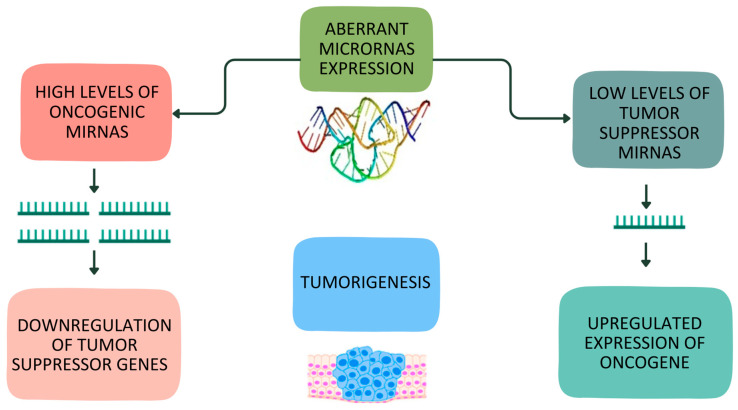
Overview of regulatory role of oncogenic and tumor suppressor miRNAs in breast cancer.

**Figure 2 ijms-24-15114-f002:**
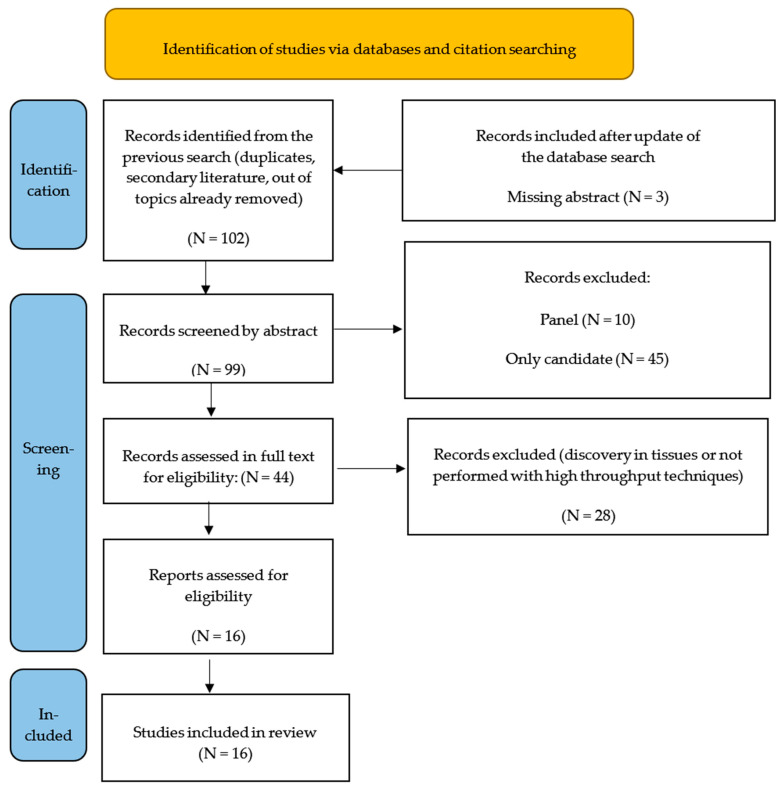
Flow chart of identification, screening, and eligibility of the included studies (identification in [13]).

**Figure 3 ijms-24-15114-f003:**
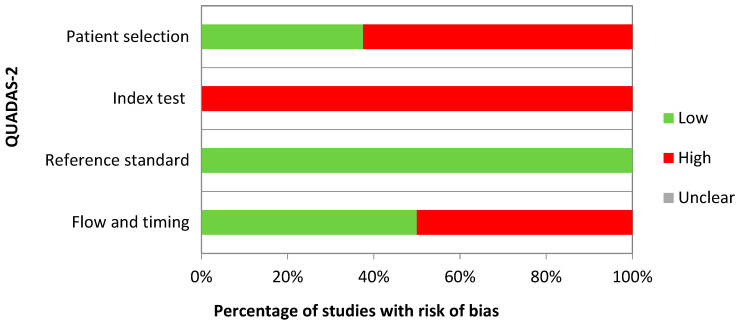
Quality assessment with the QUADAS-2 tool [18].

**Figure 4 ijms-24-15114-f004:**
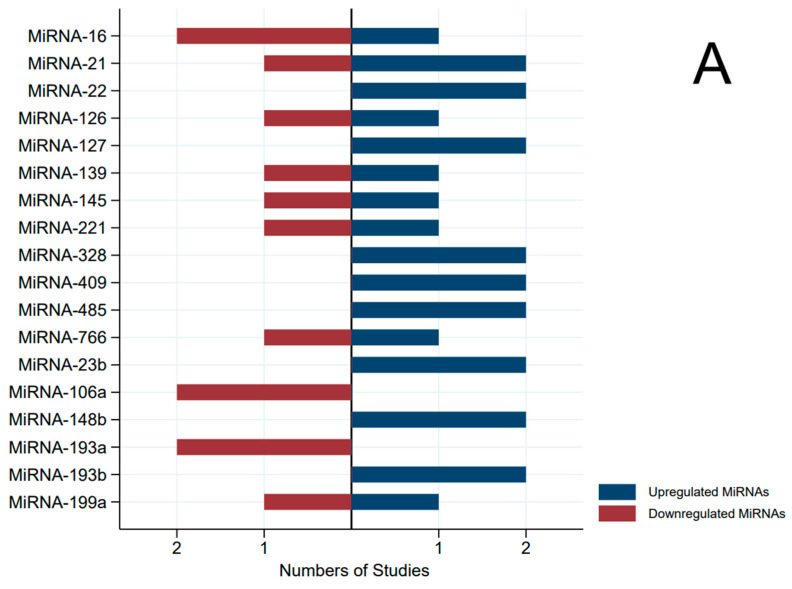
Pyramidal graph of the direction of miRNA expression (microRNA concentration in breast cancer cases versus controls) by type of specimens (only microRNAs that were analyzed in two or more independent studies). (**A**) Plasma; (**B**) serum.

**Table 1 ijms-24-15114-t001:** General features of the studies included in the systematic review on the role of microRNA in breast cancer diagnosis.

First Author, Year	Country	Specimen Source	Lab Technique	Case–Control Size	QUADAS-2 Domains with Risk of Bias	Applied Multiple Testing Correction
Schrauder MG, 2012 [20]	Germany	Blood	Geniom Biochip miRNA homo sapiens	48/57	Index test	Benjamini–Hochberg
Wu Q, 2012 [21]	China	Serum	Life Technologies SOLiD™ sequencing base miRNA expression profiling	13/10	Patient selection, Index test	Not applied
Chan M, 2013 [22]	Singapore	Serum	Agilent Human miRNA microarray	32/22	Patient selection, Index test, Flow and timing	Benjamini–Hochberg
Cuk K, 2013 [23]	Germany	Plasma	TLDA human MicroRNA Cards A v2.1 and B v2.0	10/10	Patient selection, Index test	Benjamini–Hochberg *
Ng E K, 2013 [25]	USA	Plasma	TLDA human MicroRNA Cards A v2.1 and B v2.0	5/5	Patient selection, Index test	Not applied
Godfrey AC, 2013 [24]	USA	Serum	Affimetrix GeneChip miRNA 2.0 array	205/205	Index test	Not applied
Kodahl AR, 2014 [26]	Denmark	Serum	Exiqon microRNA panel (miRCURY)	48/24	Index test	Bonferroni *
McDermott AM, 2014 [27]	Ireland	Blood	TLDA human MicroRNA Cards A v2.1 and B v2.0	10/10	Index test	Not applied
Shen J, 2014 [28]	USA	Plasma	Exiqon microRNA panel (miRCURY)	52/35	Index test, Flow and timing	Benjamini–Hochberg
Zearo S, 2014 [29]	Australia	Serum	TLDA human MicroRNA Cards A and B v3.0	39/10	Patient selection, Index test, Flow and timing	Bonferroni
Ferracin M, 2015 [31]	Italy	Plasma	Agilent Human miRNA microarray	18/18	Patient selection, Index test, Flow and timing	Not applied
Shin VY, 2015 [32]	China	Plasma	Exiqon microRNA panel (miRCURY)	5/5	Patient selection, Index test, Flow and timing	Not applied
Zhang L, 2015 [30]	China	Serum	Serum-direct multiplex qRT-PCR (SdM-qRT-PCR)	25/20	Patient selection, Index test, Flow and timing	Bonferroni, Benjamini–Hochberg
Hamam R, 2016 [33]	Saudi Arabia	Blood	Agilent Human miRNA microarray	23/9	Patient selection, Index test, Flow and timing	Benjamini–Hochberg
Jusoh A, 2021 [34]	Malaysia	Plasma	Qiagen miScript miRNA PCR Array	8/9	Index test, Flow and timing	Not applied
Záveský L, 2022 [35]	Czech Republic	Plasma	TLDA human MicroRNA Cards A v2.1 and B v2.0	7/7	Patient selection, Index test	Benjamini–Hochberg

* Cuk et al. [23] and Kodhal et al. [26] performed the adjustment for multiple comparisons but considered unadjusted *p* values for cfmiRNA selection for the validation phase.

**Table 2 ijms-24-15114-t002:** Summary of the results of the studies included in the systematic review on the role of cfmiRNAs in breast cancer diagnosis. (For cfmiRNAs analyzed in Schrauder et al. [20], Chan et al. [22], Cuk at al. [23], Kodhal et al. [26], Shen et al. [28], Zearo et al. [29], Zhang et al. [30], Hamam et al. [33], and Záveský et al. [35], adjusted *p* value were reported. Only cfmiRNAs that demonstrated statistical significance in the discovery phase have been included in the table. However, for Cuk et al. [23] and Kodhal et al. [26], non-significant adjusted *p*-values were reported since the authors considered unadjusted statistically significant *p*-values during the selection for the validation phase. About Záveský et al. [35], we decided to include all the miRNAs with a Ct-cutoff < 35, and to minimize data loss, we also added all the miRNAs that had not already been included with a Ct cut-off ≤ 40).

MIR	First Author, Year	Specimen Source	Direction	AUC	*p*-Value
1	Chan M, 2013 [22]	Serum	up		<0.001
7	Chan M, 2013 [22]	Serum	up		<0.001
16	Chan M, 2013 [22]	Serum	up		<0.001
	Ng E K, 2013 [25]	Plasma	up		
	Shin VY, 2015 [32]	Plasma	down		<0.05
	Zhang L, 2015 [30]	Serum	up		0.001
	Záveský L, 2022 [35]	Plasma	down		0.038
17	Chan M, 2013 [22]	Serum	up		0.001
	Záveský L, 2022 [35]	Plasma	down		0.017
21	Ng E K, 2013 [25]	Plasma	up		
	Ferracin M, 2015 [31]	Plasma	up		
	Shin VY, 2015 [32]	Plasma	down		<0.05
22	Shen J, 2014 [28]	Plasma	up	0.85	<0.001
	Jusoh A, 2021 [34]	Plasma	up	0.83	0.020
24	Schrauder MG, 2012 [20]	Blood	down	0.65	0.023
	Wu Q, 2012 [21]	Serum	up		
25	Wu Q, 2012 [21]	Serum	up		
	Chan M, 2013 [22]	Serum	up		<0.001
	Ng E K, 2013 [25]	Plasma	up		
28	Chan M, 2013 [22]	Serum	down		0.005
	Shen J, 2014 [28]	Plasma	up	0.85	<0.001
93	Chan M, 2013 [22]	Serum	up		<0.001
95	Chan M, 2013 [22]	Serum	up		0.023
96	Chan M, 2013 [22]	Serum	up		0.008
100	Zhang L, 2015 [30]	Serum	up	0.79	0.003
101	Zhang L, 2015 [30]	Serum	up		0.024
103	Wu Q, 2012 [21]	Serum	up		
107	Schrauder MG, 2012 [20]	Blood	down	0.68	0.041
	Chan M, 2013 [22]	Serum	up		0.013
	Kodahl AR, 2014 [26]	Serum	up		0.006
	Shen J, 2014 [28]	Plasma	up	0.87	<0.001
126	Ng E K, 2013 [25]	Plasma	down		
	Shen J, 2014 [28]	Plasma	up	0.77	<0.001
	Zearo S, 2014 [29]	Serum	up		<0.001
127	Cuk K, 2013 [23]	Plasma	up		0.459
	Shen J, 2014 [28]	Plasma	up	0.75	<0.001
128	Chan M, 2013 [22]	Serum	up		0.010
	Zhang L, 2015 [30]	Serum	up		0.039
134	Chan M, 2013 [22]	Serum	up		0.044
	Hamam R, 2016 [33]	Blood	up		0.042
136	Shen J, 2014 [28]	Plasma	up	0.87	<0.001
139	Cuk K, 2013 [23]	Plasma	down		0.320
	Kodahl AR, 2014 [26]	Serum	down		0.623
	Shen J, 2014 [28]	Plasma	up	0.79	<0.001
140	Zearo S, 2014 [29]	Serum	up		<0.001
141	Zhang L, 2015 [30]	Serum	up	0.89	0.027
142	Chan M, 2013 [22]	Serum	down		0.001
	Shen J, 2014 [28]	Plasma	up	0.82	<0.001
143	Chan M, 2013 [22]	Serum	up		<0.001
	Kodahl AR, 2014 [26]	Serum	down		0.073
	Shin VY, 2015 [32]	Plasma	down		<0.05
144	Chan M, 2013 [22]	Serum	up		<0.001
	Shen J, 2014 [28]	Plasma	down	0.94	<0.001
145	Chan M, 2013 [22]	Serum	up		0.036
	Ng E K, 2013 [25]	Plasma	down		
	Kodahl AR, 2014 [26]	Serum	down		<0.001
	Jusoh A, 2021 [34]	Plasma	up	0.82	0.040
149	Godfrey AC, 2013 [24]	Serum	up		0.030
150	Ng E K, 2013 [25]	Plasma	up		
	Hamam R, 2016 [33]	Blood	up		0.033
151	Godfrey AC, 2013 [24]	Serum	up		0.030
	Shen J, 2014 [28]	Plasma	up	0.88	<0.001
152	Shen J, 2014 [28]	Plasma	up	0.75	0.002
154	Ng E K, 2013 [25]	Plasma	up		
155	Zearo S, 2014 [29]	Serum	up		0.008
	Zhang L, 2015 [30]	Serum	up		0.017
182	Schrauder MG, 2012 [20]	Blood	down	0.71	0.008
	Chan M, 2013 [22]	Serum	up		0.009
183	Zhang L, 2015 [30]	Serum	up	0.79	0.003
184	Cuk K, 2013 [23]	Plasma	up		0.332
185	Chan M, 2013 [22]	Serum	up		<0.001
	Shin VY, 2015 [32]	Plasma	down		<0.05
186	Ng E K, 2013 [25]	Plasma	up		
	Zearo S, 2014 [29]	Serum	up		<0.001
188	Hamam R, 2016 [33]	Blood	up		0.004
190	Cuk K, 2013 [23]	Plasma	up		0.459
191	Ng E K, 2013 [25]	Plasma	up		
	Zearo S, 2014 [29]	Serum	up		<0.001
	Zhang L, 2015 [30]	Serum	up		0.018
192	Wu Q, 2012 [21]	Serum	up		
194	Wu Q, 2012 [21]	Serum	up		
	Shen J, 2014 [28]	Plasma	down	0.81	0.002
195	Chan M, 2013 [22]	Serum	up		0.007
202	Schrauder MG, 2012 [20]	Blood	up	0.72	0.020
	Zhang L, 2015 [30]	Serum	down		0.005
205	Chan M, 2013 [22]	Serum	up		0.011
206	Cuk K, 2013 [23]	Plasma	down		0.320
210	Chan M, 2013 [22]	Serum	up		0.044
	Ng E K, 2013 [25]	Plasma	up		
214	Chan M, 2013 [22]	Serum	up		<0.001
	Záveský L, 2022 [35]	Plasma	up		0.017
221	Shen J, 2014 [28]	Plasma	up	0.84	<0.001
	Shin VY, 2015 [32]	Plasma	down		<0.05
222	Wu Q, 2012 [21]	Serum	up		
	Godfrey AC, 2013 [24]	Serum	up		0.020
	Zearo S, 2014 [29]	Serum	up		<0.001
223	Wu Q, 2012 [21]	Serum	up		
	Chan M, 2013 [22]	Serum	down		<0.001
296	Chan M, 2013 [22]	Serum	up		<0.001
320	Ng E K, 2013 [25]	Plasma	down		
	Zearo S, 2014 [29]	Serum	up		<0.001
324	Ng E K, 2013 [25]	Plasma	down		
	Zhang L, 2015 [30]	Serum	up	0.88	<0.001
326	Shen J, 2014 [28]	Plasma	up	0.88	<0.001
328	Ng E K, 2013 [25]	Plasma	up		
	Shen J, 2014 [28]	Plasma	up	0.80	<0.001
330	Záveský L, 2022 [35]	Plasma	up		0.017
331	Shen J, 2014 [28]	Plasma	up	0.71	0.006
335	Schrauder MG, 2012 [20]	Blood	up	0.74	0.040
	Chan M, 2013 [22]	Serum	up		0.009
	Shen J, 2014 [28]	Plasma	up	0.73	0.006
338	Chan M, 2013 [22]	Serum	down		<0.001
339	Chan M, 2013 [22]	Serum	down		0.021
	Shen J, 2014 [28]	Plasma	up	0.76	<0.001
342	Zearo S, 2014 [29]	Serum	up		<0.001
	Shin VY, 2015 [32]	Plasma	up		<0.05
363	Chan M, 2013 [22]	Serum	up		0.003
	Godfrey AC, 2013 [24]	Serum	up		0.030
	Záveský L, 2022 [35]	Plasma	down		0.011
365	Kodahl AR, 2014 [26]	Serum	down		0.006
374	Záveský L, 2022 [35]	Plasma	down		0.022
375	Shen J, 2014 [28]	Plasma	down	0.74	0.003
378	Chan M, 2013 [22]	Serum	up		0.013
382	Shen J, 2014 [28]	Plasma	up	0.72	<0.001
409	Cuk K, 2013 [23]	Plasma	up		0.332
	Shen J, 2014 [28]	Plasma	up	0.78	<0.001
421	Chan M, 2013 [22]	Serum	up		0.009
423	Chan M, 2013 [22]	Serum	up		<0.001
	Shen J, 2014 [28]	Plasma	up	0.82	<0.001
424	Cuk K, 2013 [23]	Plasma	up		0.322
	Zhang L, 2015 [30]	Serum	up	0.86	0.002
	Hamam R, 2016 [33]	Blood	up		0.044
425	Chan M, 2013 [22]	Serum	up		0.020
	Kodahl AR, 2014 [26]	Serum	up		0.119
	Zearo S, 2014 [29]	Serum	up		<0.001
	Ferracin M, 2015 [31]	Plasma	up		
429	Wu Q, 2012 [21]	Serum	up		
451	Chan M, 2013 [22]	Serum	up		0.002
	Ng E K, 2013 [25]	Plasma	up		
454	Zearo S, 2014 [29]	Serum	up		<0.001
483	Zearo S, 2014 [29]	Serum	up		0.016
	Hamam R, 2016 [33]	Blood	up		0.038
	Záveský L, 2022 [35]	Plasma	up		0.004
484	Chan M, 2013 [22]	Serum	up		0.008
	Shen J, 2014 [28]	Plasma	up	0.84	<0.001
	Zearo S, 2014 [29]	Serum	up		<0.001
485	Ng E K, 2013 [25]	Plasma	up		
	Shen J, 2014 [28]	Plasma	up	0.87	<0.001
486	Chan M, 2013 [22]	Serum	up		<0.001
	Ng E K, 2013 [25]	Plasma	up		
	Zearo S, 2014 [29]	Serum	up		<0.001
494	Ng E K, 2013 [25]	Plasma	down		
495	Shen J, 2014 [28]	Plasma	up	0.85	<0.001
497	Schrauder MG, 2012 [20]	Blood	up	0.75	0.010
501	Chan M, 2013 [22]	Serum	up		0.023
543	Shen J, 2014 [28]	Plasma	up	0.87	<0.001
564	Schrauder MG, 2012 [20]	Blood	down	0.67	0.012
571	Cuk K, 2013 [23]	Plasma	down		0.100
574	Chan M, 2013 [22]	Serum	up		0.027
	Ng E K, 2013 [25]	Plasma	up		
	Zearo S, 2014 [29]	Serum	up		<0.001
576	Chan M, 2013 [22]	Serum	up		<0.001
584	Chan M, 2013 [22]	Serum	up		0.005
598	Chan M, 2013 [22]	Serum	up		0.020
605	Godfrey AC, 2013 [24]	Serum	down		0.050
624	Chan M, 2013 [22]	Serum	up		0.027
625	Schrauder MG, 2012 [20]	Blood	down	0.77	0.002
627	Chan M, 2013 [22]	Serum	up		0.030
629	Chan M, 2013 [22]	Serum	up		0.009
	Godfrey AC, 2013 [24]	Serum	up		0.050
652	Godfrey AC, 2013 [24]	Serum	up		0.030
660	Chan M, 2013 [22]	Serum	up		0.004
664	Chan M, 2013 [22]	Serum	down		0.050
671	Godfrey AC, 2013 [24]	Serum	up		0.010
	Záveský L, 2022 [35]	Plasma	down		0.029
718	Schrauder MG, 2012 [20]	Blood	down	0.77	0.004
744	Godfrey AC, 2013 [24]	Serum	up		0.020
760	Godfrey AC, 2013 [24]	Serum	down		0.020
762	Hamam R, 2016 [33]	Blood	up		0.042
766	Chan M, 2013 [22]	Serum	down		0.011
	Shen J, 2014 [28]	Plasma	up	0.86	<0.001
	Ferracin M, 2015 [31]	Plasma	down		
801	Cuk K, 2013 [23]	Plasma	up		0.320
874	Schrauder MG, 2012 [20]	Blood	down	0.74	0.001
	Ng E K, 2013 [25]	Plasma	down		
877	Chan M, 2013 [22]	Serum	up		0.043
922	Schrauder MG, 2012 [20]	Blood	up	0.65	0.030
1202	Hamam R, 2016 [33]	Blood	up		0.006
1207	Hamam R, 2016 [33]	Blood	up		0.020
1225	Hamam R, 2016 [33]	Blood	up		0.004
1234	Godfrey AC, 2013 [24]	Serum	down		0.030
1290	Hamam R, 2016 [33]	Blood	up		0.022
1323	Schrauder MG, 2012 [20]	Blood	up	0.69	0.040
1469	Schrauder MG, 2012 [20]	Blood	down	0.68	0.008
1471	Schrauder MG, 2012 [20]	Blood	down	0.70	0.012
1827	Godfrey AC, 2013 [24]	Serum	up		0.010
1914	Hamam R, 2016 [33]	Blood	up		0.044
1915	Schrauder MG, 2012 [20]	Blood	down	0.75	0.002
1974	Shen J, 2014 [28]	Plasma	up	0.85	<0.001
2355	Schrauder MG, 2012 [20]	Blood	down	0.73	0.004
3130	Schrauder MG, 2012 [20]	Blood	down	0.73	0.004
3136	Godfrey AC, 2013 [24]	Serum	up		0.050
3141	Hamam R, 2016 [33]	Blood	up		0.029
3156	Ferracin M, 2015 [31]	Plasma	down		
3186	Schrauder MG, 2012 [20]	Blood	down	0.75	0.002
3652	Hamam R, 2016 [33]	Blood	up		0.044
4257	Schrauder MG, 2012 [20]	Blood	up	0.65	0.040
4270	Hamam R, 2016 [33]	Blood	up		0.001
4281	Hamam R, 2016 [33]	Blood	up		0.019
4298	Hamam R, 2016 [33]	Blood	up		0.035
4306	Schrauder MG, 2012 [20]	Blood	up	0.71	0.020
	Godfrey AC, 2013 [24]	Serum	up		0.030
106a	Chan M, 2013 [22]	Serum	up		<0.001
	Ng E K, 2013 [25]	Plasma	down		
	Zhang L, 2015 [30]	Serum	up		0.018
	Záveský L, 2022 [35]	Plasma	down		0.038
106b	Schrauder MG, 2012 [20]	Blood	up	0.72	0.010
	Záveský L, 2022 [35]	Plasma	down		0.017
10a	Wu Q, 2012 [21]	Serum	up		
	Chan M, 2013 [22]	Serum	up		0.029
	Ng E K, 2013 [25]	Plasma	up		
10b	Chan M, 2013 [22]	Serum	up		<0.001
1255a	Godfrey AC, 2013 [24]	Serum	up		<0.01
125a	Wu Q, 2012 [21]	Serum	up		
	Ferracin M, 2015 [31]	Plasma	up		
125b	Zhang L, 2015 [30]	Serum	up		0.017
	Záveský L, 2022 [35]	Plasma	down		0.014
130a	Chan M, 2013 [22]	Serum	up		0.020
	Shen J, 2014 [28]	Plasma	up	0.87	<0.001
130b	Chan M, 2013 [22]	Serum	up		0.002
	Godfrey AC, 2013 [24]	Serum	up		0.030
133a	Chan M, 2013 [22]	Serum	up		<0.001
	Kodahl AR, 2014 [26]	Serum	down		0.479
	Shen J, 2014 [28]	Plasma	up	0.80	<0.001
133b	Chan M, 2013 [22]	Serum	up		<0.001
135b	Zhang L, 2015 [30]	Serum	up	0.87	<0.001
146b	Zearo S, 2014 [29]	Serum	up		<0.001
148a	Ng E K, 2013 [25]	Plasma	up		
148b	Cuk K, 2013 [23]	Plasma	up		0.320
	Shen J, 2014 [28]	Plasma	up	0.81	<0.001
15a	Kodahl AR, 2014 [26]	Serum	up		=1
15b	Chan M, 2013 [22]	Serum	up		0.003
181a	Wu Q, 2012 [21]	Serum	up		
	Chan M, 2013 [22]	Serum	down		0.023
	Godfrey AC, 2013 [24]	Serum	up		0.050
	Ferracin M, 2015 [31]	Plasma	down		
	Zhang L, 2015 [30]	Serum	up	0.86	<0.001
181b	Wu Q, 2012 [21]	Serum	up		
181c	Chan M, 2013 [22]	Serum	down		0.038
18a	Chan M, 2013 [22]	Serum	up		0.004
	Godfrey AC, 2013 [24]	Serum	up		0.040
	Kodahl AR, 2014 [26]	Serum	up		0.007
18b	Chan M, 2013 [22]	Serum	up		0.002
	Godfrey AC, 2013 [24]	Serum	down		0.040
193a	Schrauder MG, 2012 [20]	Blood	down	0.79	<0.001
	Cuk K, 2013 [23]	Plasma	down		0.320
	Ng E K, 2013 [25]	Plasma	down		
193b	Wu Q, 2012 [21]	Serum	up		
	Ng E K, 2013 [25]	Plasma	up		
	Zhang L, 2015 [30]	Serum	up	0.80	0.002
	Záveský L, 2022 [35]	Plasma	up		0.017
196b	Záveský L, 2022 [35]	Plasma	down		0.041
199a	Chan M, 2013 [22]	Serum	down		0.013
	Shen J, 2014 [28]	Plasma	up	0.84	<0.001
	Shin VY, 2015 [32]	Plasma	down		<0.05
	Zhang L, 2015 [30]	Serum	up	0.84	0.001
19a	Chan M, 2013 [22]	Serum	up		0.016
	Záveský L, 2022 [35]	Plasma	down		0.038
200b	Wu Q, 2012 [21]	Serum	up		
200c	Wu Q, 2012 [21]	Serum	up		
	Ng E K, 2013 [25]	Plasma	up		
20a	Chan M, 2013 [22]	Serum	up		<0.001
	Záveský L, 2022 [35]	Plasma	down		0.017
20b	Chan M, 2013 [22]	Serum	up		0.001
	Záveský L, 2022 [35]	Plasma	down		0.011
23a	Wu Q, 2012 [21]	Serum	up		
23b	Wu Q, 2012 [21]	Serum	up		
	Shen J, 2014 [28]	Plasma	up	0.76	0.009
	Shin VY, 2015 [32]	Plasma	up		<0.05
26a	Wu Q, 2012 [21]	Serum	up		
26b	Chan M, 2013 [22]	Serum	down		0.005
	Záveský L, 2022 [35]	Plasma	down		0.011
27a	Wu Q, 2012 [21]	Serum	up		
	Ng E K, 2013 [25]	Plasma	up		
27b	Wu Q, 2012 [21]	Serum	up		
	Jusoh A, 2021 [34]	Plasma	up	0.82	0.010
29a	Wu Q, 2012 [21]	Serum	up		
	Zearo S, 2014 [29]	Serum	up		<0.001
	Zhang L, 2015 [30]	Serum	up		0.029
29b	Wu Q, 2012 [21]	Serum	up		
29c	Wu Q, 2012 [21]	Serum	up		
	Zhang L, 2015 [30]	Serum	up	0.81	0.001
30a	Chan M, 2013 [22]	Serum	up		0.029
30b	Chan M, 2013 [22]	Serum	down		0.027
	Shen J, 2014 [28]	Plasma	up	0.76	<0.001
30c	Shen J, 2014 [28]	Plasma	up	0.77	<0.001
30d	Chan M, 2013 [22]	Serum	up		0.008
30e	Wu Q, 2012 [21]	Serum	up		
320a	Wu Q, 2012 [21]	Serum	up		
	Chan M, 2013 [22]	Serum	up		<0.001
	Ferracin M, 2015 [31]	Plasma	up		
320b	Chan M, 2013 [22]	Serum	up		<0.001
320d	Godfrey AC, 2013 [24]	Serum	up		0.040
33a	Shen J, 2014 [28]	Plasma	up	0.79	<0.001
34a	Hamam R, 2016 [33]	Blood	up		0.044
374a	Shen J, 2014 [28]	Plasma	up	0.75	0.004
374b	Chan M, 2013 [22]	Serum	down		0.007
376a	Cuk K, 2013 [23]	Plasma	up		0.386
376c	Cuk K, 2013 [23]	Plasma	up		0.224
449b	Zhang L, 2015 [30]	Serum	up	0.89	<0.001
516b	Schrauder MG, 2012 [20]	Blood	up	0.67	0.030
	Zhang L, 2015 [30]	Serum	up		0.038
519a	Cuk K, 2013 [23]	Plasma	down		0.407
519c	Zhang L, 2015 [30]	Serum	up	0.85	0.003
520c	Zhang L, 2015 [30]	Serum	up	0.80	0.003
526a	Schrauder MG, 2012 [20]	Blood	down	0.72	0.013
526b	Cuk K, 2013 [23]	Plasma	down		0.386
548b	Záveský L, 2022 [35]	Plasma	up		0.001
548c	Záveský L, 2022 [35]	Plasma	up		0.035
548d	Godfrey AC, 2013 [24]	Serum	down		0.010
	Záveský L, 2022 [35]	Plasma	up		0.002
551a	Chan M, 2013 [22]	Serum	down		0.002
642b	Hamam R, 2016 [33]	Blood	up		0.020
92a	Wu Q, 2012 [21]	Serum	up		
	Chan M, 2013 [22]	Serum	up		<0.001
	Shin VY, 2015 [32]	Plasma	up		<0.05
92b	Chan M, 2013 [22]	Serum	up		0.003
99b	Shen J, 2014 [28]	Plasma	up	0.81	<0.001
let7a	Schrauder MG, 2012 [20]	Blood	up	0.65	0.030
let-7a	Chan M, 2013 [22]	Serum	up		0.005
let-7b	Chan M, 2013 [22]	Serum	up		<0.001
	Zearo S, 2014 [29]	Serum	up		<0.001
	Záveský L, 2022 [35]	Plasma	down		0.026
let-7c	Chan M, 2013 [22]	Serum	up		0.009
	Záveský L, 2022 [35]	Plasma	down		0.038
let-7d	Shen J, 2014 [28]	Plasma	up	0.89	<0.001
let-7f	Chan M, 2013 [22]	Serum	up		0.016
	Shen J, 2014 [28]	Plasma	up	0.81	<0.001
let-7g	Chan M, 2013 [22]	Serum	up		0.002
	Ng E K, 2013 [25]	Plasma	up		
let-7i	Chan M, 2013 [22]	Serum	up		<0.001
U6 snRNA	Záveský L, 2022 [35]	Plasma	up		0.004

## Data Availability

Not applicable.

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
