# Peer review of "Identifying MicroRNAs Suitable for Detection of Breast Cancer: A Systematic Review of Discovery Phases Studies on MicroRNA Expression Profiles"

_ijms, 2023, doi:10.3390/ijms242015114_

Round 1
Reviewer 1 Report
The review by Padroni et al., "Identifying miRNAs suitable for detection of breast cancer: a systematic review of discovery phases studies on miRNA expression profiles" elegantly describes how CfmiRNAs can be used as diagnostic and prognostic biomarkers for breast cancer patients. They have shortlisted several cfmiRNAs, including MIR16, MIR191, MIR484, MIR106a, and MIR193b, that showed significant promise to be considered as diagnostic biomarkers with proper validation.
This is a well-debated article that summarizes 17 different studies to shortlist these CfmiRNAs. The authors also articulated the limitations and challenges the current field is facing in shortlisting biomarkers due to inconsistent study designs. They have done an excellent job, and this review would be a great way to provide further guidance for scientists who want to dive into biomarker research field to make significant discoveries.
Below I provide comments to improve this review article.
1. Page 4: Can you elaborate on “risk of bias was found in many studies for patient selection, index testing and flow and timing”.
2. The authors should include a simple schema to show what cfmiRNAs are and how they can be deregulated in breast cancer.
3. The authors should provide a brief/elaborate table summarizing the optimal way to characterize these cfmiRNAs. Different parameters should be included- sample collection, preservation, laboratory methodologies, cfmiRNAs measurement and normalization, and cut-off values. It can be a very effective guidelines for future studies.
4. The authors should include another paragraph emphasizing what could be done to enrich the field.
Provide few ideas-
a) More studies should be done on Black and Hispanic populations,
b) Studies should include a specific age group or breast cancer patients with other medical conditions like Primary Ovarian Insufficiency (POI) or premature menopause.
Moderate editing of English language required.
Author Response
I would like to express our gratitude to both of the reviewers for their valuable comments, which have greatly contributed to the enhancement of our paper.
|
Reviewer 1 |
Answer |
|
1. Page 4: Can you elaborate on “risk of bias was found in many studies for patient selection, index testing and flow and timing”. |
We provided a more comprehensive explanation of the sentence in question within the Results section |
|
2. The authors should include a simple schema to show what cfmiRNAs are and how they can be deregulated in breast cancer. |
Thank you very much for your suggestion. We included in the introduction a section on miRNA role in breast cancer and a simple figure (figure 1). |
|
3. The authors should provide a brief/elaborate table summarizing the optimal way to characterize these cfmiRNAs. Different parameters should be included- sample collection, preservation, laboratory methodologies, cfmiRNAs measurement and normalization, and cut-off values. It can be a very effective guidelines for future studies. |
Thank you very much for your suggestion. The endeavour to establish a checklist for describe In a comprehensive manner, biomarker studies in articles was undertaken by Gallo et al. in the STROBE-ME initiative. Hence, recommending the consistent adoption of this easily usable, standardized, and universally accepted checklist, often considered the gold standard for observational studies reports, appears most fitting. |
|
4. The authors should include another paragraph emphasizing what could be done to enrich the field. |
Thank you very much for your suggestion. We included a new paragraph in the conclusion section. |
|
Moderate editing of English language required. |
The English text has been thoroughly revised. |
|
|
|

Reviewer 2 Report
This systematic review by Padroni et al examines the role of circulation miRNAs in the context of breast cancer detection. A literature search revealed 17 eligible publications, which were then screened for potential diagnostic biomarkers.
Major points:
1. The authors note that the studies had a high risk of bias and present an overall graph with corresponding percentages of studies (Figure 3). Since this is one of the main results of the review, the respective analysis should be described in much more detail and results for the individual studies should be presented. In addition, “low risk of applicability” should presumably read “low concern of applicability” in Figure 3 and the Results section.
2. Did the authors exclude studies with genome-wide data which were not part of a discovery phase? E.g. studies not divided in a discovery and validation phase or accessing genome-wide data also in the validation phase? All non-candidate studies fulfilling the remaining inclusion criteria should be included in this review.
3. Figure 2 and Tables 1 and 2:
a. The selection of miRNAs remains in part unclear, including whether a filtering based on unadjusted or adjusted p-values was performed. According to the discussion “[…] not all the analyses were adjusted for multiple testing”. Please describe the selection in more detail.
b. If very different criteria were applied, the given presentation may be problematic, and harmonization should be sought as much as possible. Moreover, the different platforms used in the various studies seem to overlap only partially (e.g. some arrays seem to assess only about 1000 mature human miRNAs). I.e., variants selected as candidates in one study may not be measured in another study, affecting the interpretation of results.
c. Tables 1 and 2 are very long and therefore difficult to read. My suggestion would be to delete the MiR column in Table 1 - the miRNAs are listed in Table 2. Table 2 should also include confidence intervals and p-values if available. To improve readability, Table 2 could be replaced/complemented by a supplementary table with one row per miRNA, listing the effect sizes etc. of the different studies in columns regardless of whether a miRNA was selected as a candidate or not. This would also reveal whether a certain miRNAs was assessed/measured in a study or not.
4. In the discussion, Padroni et al. state that “Consequently, the signals that are then chosen for replication must show a high statistical significance and be rigorously corrected for multiple testing. […] A trivial exemplification of lack of standardization in analyses is that not all the analyses were adjusted for multiple testing.” Multiple testing correction is required for validation, but not necessarily for discovery. Another point is that winner’s curse bias may be induced if a p-value cutoff is applied for the selection of validation candidates. If a major reason for the lack of standardization in analyses is related to multiple testing, could the authors apply the identical correction procedure to all studies to achieve better comparability?
5. Five miRNAs (MIR16, MIR191, MIR484, MIR106a, and MIR193b) are described as good candidates for validation. However, some of the study effect sizes are very different and even in opposite directions. What is the rationale for proposing these candidates? Have these variants been considered in published validation phases/studies?
Minor points:
1. Some numbers do not match in the first paragraph of the results section:
a. “…we excluded 202 records…” should presumably read “… we excluded 206 records…”
b. According to the Text and Figure 1, n=3 of the 102 records were excluded because no English abstract was available, resulting in n=99 records. To finally end up with n=17 studies, another n=82 seem to have been excluded. In contrast, the text notes that “…an additional group of 85 publications were excluded…”. It seems that the n=3 studies mentioned above were considered here. Please clarify.
2. Please re-check the platform names given in Table 1 and give version numbers for arrays. E.g. according to Schrauder et al. (2012) a “ Geniom Biochip miRNA homo sapiens” was applied and Cuk et al. (2013) used TLDA human MicroRNA Cards A v2.1 & B v2.0.
Moderate editing of English language is required.
Author Response
|
Reviewer 2 |
|
|
Major points: |
|
|
1. The authors note that the studies had a high risk of bias and present an overall graph with corresponding percentages of studies (Figure 3). Since this is one of the main results of the review, the respective analysis should be described in much more detail and results for the individual studies should be presented. In addition, “low risk of applicability” should presumably read “low concern of applicability” in Figure 3 and the Results section. |
We have made the requested changes in the text
|
|
Did the authors exclude studies with genome-wide data which were not part of a discovery phase? E.g. studies not divided in a discovery and validation phase or accessing genome-wide data also in the validation phase? All non-candidate studies fulfilling the remaining inclusion criteria should be included in this review. |
We did not exclude genome-wide studies, nor did we exclude studies that were not divided into discovery and validation. |
|
Figure 2 and Tables 1 and 2: |
|
|
The selection of miRNAs remains in part unclear, including whether a filtering based on unadjusted or adjusted p-values was performed. |
We did not make any selection of miRNAs. We included all the miRNAs listed in the 17 papers in Table 2. Furthermore, in the pyramidal graphs, we only included miRNAs that were analyzed in two or more studies. |
|
According to the discussion “[…] not all the analyses were adjusted for multiple testing”. Please describe the selection in more detail. |
We did not make any multiple testing adjustments, but this sentence referred to the original statistical analyses of the papers. Thank you for pointing out that the sentence was unclear. We have now completely rephrased this section of the discussion to be more clear. |
|
If very different criteria were applied, the given presentation may be problematic, and harmonization should be sought as much as possible. Moreover, the different platforms used in the various studies seem to overlap only partially (e.g. some arrays seem to assess only about 1000 mature human miRNAs). I.e., variants selected as candidates in one study may not be measured in another study, affecting the interpretation of results. |
This is one of the problems of miRNA studies that this review aims to highlight. In theory, even different platforms should guarantee the identification of the most promising biomarkers, but this factor certainly greatly increases the variability of the results. We have tried to better emphasize this concept in the text. |
|
Tables 1 and 2 are very long and therefore difficult to read. My suggestion would be to delete the MiR column in Table 1 - the miRNAs are listed in Table 2. Table 2 should also include confidence intervals and p-values if available. To improve readability, Table 2 could be replaced/complemented by a supplementary table with one row per miRNA, listing the effect sizes etc. of the different studies in columns regardless of whether a miRNA was selected as a candidate or not. This would also reveal whether certain miRNAs were assessed/measured in a study or not |
Thank you very much for your suggestion. We rearranged tables 1 and 2 as per your suggestion |
|
In the discussion, Padroni et al. state that “Consequently, the signals that are then chosen for replication must show a high statistical significance and be rigorously corrected for multiple testing. […] A trivial exemplification of lack of standardization in analyses is that not all the analyses were adjusted for multiple testing.” Multiple testing correction is required for validation, but not necessarily for discovery. Another point is that winner’s curse bias may be induced if a p-value cutoff is applied for the selection of validation candidates. If a major reason for the lack of standardization in analyses is related to multiple testing, could the authors apply the identical correction procedure to all studies to achieve better comparability? |
Thank you very much for your suggestion. we agree with you that adjusting for multiple testing is not required in the discovery phase and we have modified the text accordingly. Concerning the potential for the winner’s curse bias, we acknowledge that implementing a p-value cutoff for candidate selection could introduce inflated effect sizes, thereby potentially distorting results. In light of this, it is worth noting that mitigating the winner’s curse bias is feasible by adopting a more stringent significance threshold for variant selection. However, it is essential to strike a careful balance between addressing this bias and potential ramifications, such as diminished statistical power due to the selection of a reduced number of variants.
|
|
5. Five miRNAs (MIR16, MIR191, MIR484, MIR106a, and MIR193b) are described as good candidates for validation. However, some of the study effect sizes are very different and even in opposite directions. What is the rationale for proposing these candidates? Have these variants been considered in published validation phases/studies? |
We are aware of and have highlighted the limitations of these results. The selection of the 5 miRNAs was based on their candidacy for validation in at least two studies using the same specimen source. |
|
Some numbers do not match in the first paragraph of the results section: |
We fixed the problem. Thank you for noticing it. |
|
According to the Text and Figure 1, n=3 of the 102 records were excluded because no English abstract was available, resulting in n=99 records. To finally end up with n=17 studies, another n=82 seem to have been excluded. In contrast, the text notes that “…an additional group of 85 publications were excluded…”. It seems that the n=3 studies mentioned above were considered here. Please clarify. |
We fixed the problem. Thank you for noticing it. |
|
Please re-check the platform names given in Table 1 and give version numbers for arrays. E.g. according to Schrauder et al. (2012) a “ Geniom Biochip miRNA homo sapiens” was applied and Cuk et al. (2013) used TLDA human MicroRNA Cards A v2.1 & B v2.0. |
We fixed the problem. Thank you for noticing it. |
|
Moderate editing of English language is required |
The English text has been thoroughly revised. |

Round 2
Reviewer 2 Report
The authors have satisfactorily answered some of the criticisms, however the following points did remain:
Major points:
1. The presentation of the main result, the QUADAS-2 analysis, is still brief. The percentages shown in Figure 4 should be stated in the text and a column should be added to Table 1, indicating whether a risk of bias was observed for a study and if yes, in which domain. E.g. “Patient selection, Index test” in case of a risk of bias for these two, but not the other two domains. To save space, case and control sizes could be combined in one column.
2. In the results section “…index testing relates to the standard of reference used…” should probably read “…. reference standard relates to…”. It should be clearly described what the reference standard is in this analysis.
3. In their response, the authors clarify that all genome-wide studies satisfying the filter criteria were in principle considered, even if the data were not part of a discovery study. However, this is not clearly stated in the manuscript [e.g. inclusion criterion 3): “incorporation of a discovery phase that used high throughput techniques”] and should be revised.
4. The authors rephrased the second part of the discussion, yet deleted the part on multiple testing. The reader should be made aware of that some studies did not use multiple testing and that this was ignored in the miRNA selection. The studies with/without multiple testing should be marked in Table 1 or cited in the text.
5. The fold changes in Table 2 are partly negative, which is impossible. Please correct or revise the column heading. In addition, to help the reader to better judge the results, confidence intervals should be included in Table 2 when available.
6. Proposing the five candidate miRNAs (MIR16, MIR191, MIR484, MIR106a, and MIR193b) just based on their candidacy in at least two studies with identical specimen source seems not very convincing, given the reported risk of bias and the in part extremely different effect sizes. Of course, additional factors such as ethnicity could affect the results, but these limitations should be discussed and the formulation as “…probably good candidates…” should be at least weakened.
Minor points:
1. “Low risk of applicability” should presumably read “low concern of applicability” in Figure 4 and the Results section. In addition, Figure 4 the label for the last column is missing (probably “flow and timing”). Moreover, in “… some studies failed to mention whether the threshold was prespecified…” in the results section, “the threshold” should be replaced by “a threshold”.
2. In the Introduction, there seems to be one word missing at the end of the following sentence or there is an extra “to”: “In particular, cfmiRNAs have been extensive investigated as a diagnostic biomarkers, other than as a biomarkers of for prognosis and of therapy response to.”
Minor editing of English language required.
Author Response
Responses to the reviewer comments
|
Reviewer 2 |
Answer |
|
1. The presentation of the main result, the QUADAS-2 analysis, is still brief. The percentages shown in Figure 4 should be stated in the text and a column should be added to Table 1, indicating whether a risk of bias was observed for a study and if yes, in which domain. To save space, case and control sizes could be combined in one column. |
We included a new column in table 1 and we added the percents of high risk of bias in the text |
|
2. In the results section “…index testing relates to the standard of reference used…” should probably read “…. reference standard relates to…”. It should be clearly described what the reference standard is in this analysis. |
Unfortunately, there is not yet a reference standard for circulating miRNAs in breast cancer. The reference is based on the ROC curve to obtain the optimal diagnostic threshold, so there is no unified diagnostic threshold standard. This point was clarified in the text. |
|
3. In their response, the authors clarify that all genome-wide studies satisfying the filter criteria were in principle considered, even if the data were not part of a discovery study. However, this is not clearly stated in the manuscript [e.g. inclusion criterion 3): “incorporation of a discovery phase that used high throughput techniques”] and should be revised. |
We have clarified the inclusion criteria, as suggested. |
|
4. The authors rephrased the second part of the discussion, yet deleted the part on multiple testing. The reader should be made aware of that some studies did not use multiple testing and that this was ignored in the miRNA selection. The studies with/without multiple testing should be marked in Table 1 or cited in the text. |
Thank you for your suggestion. We included a new paragraph in the results and conclusion section on the topic of multiple testing and included a new column in table 1 as suggested. |
|
5. The fold changes in Table 2 are partly negative, which is impossible. Please correct or revise the column heading. In addition, to help the reader to better judge the results, confidence intervals should be included in Table 2 when available. |
We revised the negative fold changes. Unfortunately there are very few confidence intervals, then we cannot add a new column. |
|
6. Proposing the five candidate miRNAs (MIR16, MIR191, MIR484, MIR106a, and MIR193b) just based on their candidacy in at least two studies with identical specimen source seems not very convincing, given the reported risk of bias and the in part extremely different effect sizes. Of course, additional factors such as ethnicity could affect the results, but these limitations should be discussed and the formulation as “…probably good candidates…” should be at least weakened. |
We have reformulated the conclusions as suggested. |
|
1. “Low risk of applicability” should presumably read “low concern of applicability” in Figure 4 and the Results section. In addition, Figure 4 the label for the last column is missing (probably “flow and timing”). Moreover, in “… some studies failed to mention whether the threshold was prespecified…” in the results section, “the threshold” should be replaced by “a threshold”. |
We fixed the problem. Thank you for noticing it. |
|
2. In the Introduction, there seems to be one word missing at the end of the following sentence or there is an extra “to”: “In particular, cfmiRNAs have been extensive investigated as a diagnostic biomarkers, other than as a biomarkers of for prognosis and of therapy response to.” |
We fixed the problem. Thank you for noticing it. |

Round 3
Reviewer 2 Report
I thank the authors for their detailed description and their additions to the manuscript. Yet, the description of the reference standard still remains somewhat vague (“…based on the ROC curve to obtain the optimal diagnostic threshold…”). Is this concerned with the method used to determine the presence or absence of breast cancer in the different studies? Otherwise, AUCs are only listed for 5 publications in Table 2; at least in a part of the remaining publications no ROC analysis seems to have been performed.
I suggest deleting the newly inserted paragraph in the discussion again (or to rephrase it). It is now clear from the last column of Table 1 that correction for multiple testing was only applied in some studies. As mentioned in the first round, multiple testing correction may not be necessary in a discovery phase. Therefore, standardization in this regard cannot be generally required of the studies (in contrast of reporting the number of test performed and unadjusted p-values, which would enable Padroni et al to conduct a Bonferroni correction for all studies, and thus achieve standardization). In addition, winner’s curse also occurs independent of multiple testing, i.e. when applying a cutoff to unadjusted p-values, and a winner’s curse correction method could be applied to at correct for inflated effect sizes.
Please also carefully review the results presented in Table 2. Some information should be added, and few errors seem to have crept in during the transcription of the results. E.g., two results are listed for miR-22, Shen et al. with FC of 29.228 and 43.356, whereas Table 1 in Shen et al reports FC of 2.9228 for miR-22 and 4.3356 for miR-22* (the negative FC in Shen et al. presumably have to be interpreted such that –x should be correctly noted as 1/x). In addition, some of the fold changes in Table 2 with upward direction are < 1 and some with downward direction >1. This should be harmonized and the direction clarified in the column heading or a footnote (e.g. “Fold change (case/control)”). (delta)deltaCT/CP-values given in Cuk et al. and Kodahl et al. can and should be converted to fold changes. Schrauder et al. and Hamam et al. report adjusted p-values, but Table 2 lists no p-values or unadjusted p-values; for Ng et al several miRNAs are reported which seem to be missing from the cited publication (miRs 126, 145, 320, 324, 494, 874, 106a, and 193a). The results presented for McDermott et al. and Ferracin et al. are from validation of selected candidates, not from discovery analysis using high throughput techniques.
As for the publication of Holubekova et al (2020), it appears that no high throughput techniques were used, but only n=12 candidates selected from literature were tested. In addition, no multiple testing correction seem to be reported and pilot testing was only conducted in 15 vs 15 samples, in contrast to the data in Table 1. Please revise whether this publication satisfies the inclusion criteria.
Author Response
Responses to the reviewer comments
|
Reviewer 2 |
Answer |
|
Yet, the description of the reference standard still remains somewhat vague (“…based on the ROC curve to obtain the optimal diagnostic threshold…”). Is this concerned with the method used to determine the presence or absence of breast cancer in the different studies? |
The QUADAS 2 reference standard domain assesses the accuracy of disease status classification. Test accuracy estimates assume a reference standard with 100% sensitivity and specificity. In this review, we verified whether breast cancer diagnoses were histologically confirmed. Thank you for pointing out the lack of clarity in the sentence. We have rewritten it, and we hope it is now much clearer. |
|
I suggest deleting the newly inserted paragraph in the discussion again (or to rephrase it). It is now clear from the last column of Table 1 that correction for multiple testing was only applied in some studies. As mentioned in the first round, multiple testing correction may not be necessary in a discovery phase. Therefore, standardization in this regard cannot be generally required of the studies (in contrast of reporting the number of test performed and unadjusted p-values, which would enable Padroni et al to conduct a Bonferroni correction for all studies, and thus achieve standardization).
|
As suggested, we have removed the newly inserted paragraph because the last column of Table 1 suffices to highlight the issue of multiple testing correction. We cannot apply the traditional Bonferroni method to correct for multiple comparisons in studies without multiple test correction. This is because the Bonferroni method is overly conservative, and its stringent criteria for guarding against false positives would result in the exclusion of many potentially significant findings (VanderWeele TJ, AJE 2019 Mar 1;188(3):617-618.). The required p-value threshold for the number of comparisons in miRNA studies is approximately <0.0004. In fact, the majority of studies included in this review has opted for the False Discovery Rate (FDR) method. Nevertheless, we are unable to perform the FDR method for the studies included in the review, due to the unavailability of the complete list of p-values for all tested miRNAs. For these reasons, we believe that including a column indicating the status of multiple testing correction is the best approach to warn the reader of the lack of standardization.
In addition to adding a column for multiple testing correction, we found it useful to include one describing the criteria for selecting miRNAs for validation. In some cases, miRNA selection was not solely based on p-values, and adjusted p-values were not always the sole criteria, even after applying multiple testing correction. |
|
In addition, winner’s curse also occurs independent of multiple testing, i.e. when applying a cutoff to unadjusted p-values, and a winner’s curse correction method could be applied to at correct for inflated effect sizes. |
Furthermore, we agree with the reviewer that Winner’s Curse could exaggerate the miRNA association estimates in the sample in which these associations were discovered. This isn’t just a phenomenon related to miRNA but this is a common phenomenon in all the molecular epidemiology studies. Anyway, the present study is a qualitative review not a quantitative meta-analysis, and we cannot control for Winner's Curse bias. |
|
Please also carefully review the results presented in Table 2. Some information should be added, and few errors seem to have crept in during the transcription of the results. E.g., two results are listed for miR-22, Shen et al. with FC of 29.228 and 43.356, |
We carefully reviewed the results of table 1 and 2. We reported now the correct result for Shen et al., thank you for noticing it. |
|
whereas Table 1 in Shen et al reports FC of 2.9228 for miR-22 and 4.3356 for miR-22* (the negative FC in Shen et al. presumably have to be interpreted such that –x should be correctly noted as 1/x). In addition, some of the fold changes in Table 2 with upward direction are < 1 and some with downward direction >1. This should be harmonized and the direction clarified in the column heading or a footnote (e.g. “Fold change (case/control)”). (delta) deltaCT/CP-values given in Cuk et al. and Kodahl et al. can and should be converted to fold changes. |
We would like to express our sincere gratitude for bringing up this issue which made us reflect more deeply on the opportunity of presenting the fold changes of the miRNAs in the table 2.
The calculation of fold changes varies among different articles: some are performed on normalized data, while others are not; some use ratios, while others employ the standard 2^-ΔΔCt formula. Additionally, some articles do not specify their calculation methods. In detail:
Schrauder MG, 2012, Shin VY, 2015 and Jusoh A, 2021 used the 2^-ΔΔCt with different standardization
Chan M, 2013; Used the log2 2^-ΔΔCt
Hamam R, 2016 Probably used the 2^-ΔΔCt formula but it is not declared
Shen J, 2014 Zearo S, 2014 Ng E K, 2013 and Zhang L, 2015 calculated fold change probably with a log transformed 2^-ΔΔCt formula but not clear
Cuk K, 2013 and Kodahl AR, 2014 Present only ΔC we could calculate 2^-ΔΔCt
Wu Q, 2012 and Ferracin M, 2015 Used the ratio among copy number in cases and controls
Godfrey AC, 2013 Percent change
Záveský L, 2022 Fold difference
McDermott AM, 2014 Fold change not calculated and not possible to calculate it.
In light of these reflections, we realized that inserting a fold change column in the table, obtained through non-homogeneous methods, would not provide meaningful information |
|
Schrauder et al. and Hamam et al. report adjusted p-values, but Table 2 lists no p-values or unadjusted p-values; for Ng et al several miRNAs are reported which seem to be missing from the cited publication (miRs 126, 145, 320, 324, 494, 874, 106a, and 193a). |
We have undertaken revisions to Table 2, specifically addressing the P-values for Schrauder et al. and Hamam et al., incorporating the necessary amendments. Additionally, we have included the previously omitted miRNAs as reported in the study conducted by Ng et al. |
|
The results presented for McDermott et al. and Ferracin et al. are from validation of selected candidates, not from discovery analysis using high throughput techniques. |
We have meticulously reviewed the work of McDermott et al., and finally we agree that the numbers in the table are not related to the discovery analysis. Consequently, we have made the necessary amendments to rectify this oversight. We also rectified the values related to the paper by Ferracin et al.. |
|
As for the publication of Holubekova et al (2020), it appears that no high throughput techniques were used, but only n=12 candidates selected from literature were tested. In addition, no multiple testing correction seem to be reported and pilot testing was only conducted in 15 vs 15 samples, in contrast to the data in Table 1. |
After a carefull reading of Holubekova et al and a discussion among authors we agreed with the reviewer and we excluded this paper from the review. |

Round 4
Reviewer 2 Report
I generally agree with the changes made. The addition of an additional column describing the selection method is very welcome and will help the reader to better understand and interprete the data presented. Yet, a part of the criteria listed refer to the selection of candidates for validation, while others describe the (discovery) results included in Table 2. Concerning the former, not all of the 16 publications included a validation step, validated only scores/multi-miRNA models, chose the number of candidates based on the sample size available for validation or used additional criteria not mentioned in the newly added column (e.g. analysis of BC tissue, samples from other cancer entities, or results from literature). Since the authors focus on the discovery part of the studies, a column describing the selection methods for validation may not be essential.
In contrast, a description of the results included in Table 2 will be very valuable tor the reader. Here (or in footnotes to Table 2), differences between the discovery results described in the publications (applying the listed criteria) and Table 2 should be clearly described: E.g., Schrauder et al. report n=59 miRNAs with adj. p < 5%, but only the n=25 top hits are shown in Table 2 since data for the remaining 24 are not included in their publication. Please review the criteria carefully (e.g., McDermott do not seem to apply a p-value cutoff, but a rank ordering based on ANN models and Monte Carlo simulations). It should be clearly stated whether only upregulated or up- and down-regulated miRNAs were considered (e.g. using FC>2 or <1/2 in the latter case) and p<5% or p<=5% was used (likewise for FC>2 or >=2; some miRNAs have p=0.05 in Table 2).
In this context, Table 2 should be updated: For example, no miRNAs are listed for McDermott et al. and only n=3 for Jusoh et al., whereas the published discovery results comprise 76 and 40 miRNAs, respectively. In addition, it should be mentioned if samples of non-BC cancer types or BC tissue (e.g. Ferracin et al) were used for selection or – as this may be more within the scope of the manuscript - miRNA selection/data presented should be based only on the published results for BC blood/plasma/serum samples vs controls if possible (the same applies to Shin et al, where TNBC vs non-TNBC results were used for selection & Table 2 contains data of the TNBC vs normal-comparison). Furthermore, Zavesky et al used two different Ct-cutoffs (Ct<35 and Ct<=40, not Ct>40) and the results of both were merged in Table 2 without indication and explanation how the data was combined. The latter also applies to the nondiscrimination between mature miRNAs originating from the opposite arms of the same precursor miRNA (i.e., nondiscrimination between suffixes “-3p”, “-5p”, “*” or no suffix). This should be mentioned e.g. in a footnote or the text, including how data for a study was merged in this case.
Minor points:
11. According to Table 1, N=6 studies used serum and N=3 blood, but N=7 and 2 are mentioned on page 4.
22. Percentages on page 5 referring to Figure 3 should be updated (presumably 62.5%, 100% and 50%).
33. 50 cases/controls are listed for Wu et al. in Table 1, whereas "13 BC patients vs. 10 healthy controls” are described for the discovery part in the publication.
44. Godfrey et al.: p-value for 1255a in Table 2 should presumably read p<0.01 instead of <0.001
55. Please recheck platform names and versions in Table 1 (e.g., Affymetrix GeneChip miRNA 2.0 array in Godfrey et al, Version v3.0, not of 2.1/2.0 of TaqMan Array Human MicroRNA Cards A and B in Zearo et al)
66. Please replace p>1 with p=1 in Table 2
77. “snRNA” at the end of Table 2 should presumably read U6 “sRNA”
88. miRNA 451 in Zavesky et al (Table 1) seems to be from mouse, please indicate this e.g. in a footnote
Author Response
Responses to the reviewer comments
|
Reviewer 2 |
Answer |
|
Yet, a part of the criteria listed refer to the selection of candidates for validation, while others describe the (discovery) results included in Table 2. Concerning the former, not all of the 16 publications included a validation step, validated only scores/multi-miRNA models, chose the number of candidates based on the sample size available for validation or used additional criteria not mentioned in the newly added column (e.g. analysis of BC tissue, samples from other cancer entities, or results from literature). Since the authors focus on the discovery part of the studies, a column describing the selection methods for validation may not be essential. |
We thought that a column explaining the different methods used by the authors to select the miRNAs for validation could have been useful to the reader. However, we realized it might be misleading, so to avoid any confusion, we decided to remove the column from Table 1. |
|
In contrast, a description of the results included in Table 2 will be very valuable tor the reader. Here (or in footnotes to Table 2), differences between the discovery results described in the publications (applying the listed criteria) and Table 2 should be clearly described: E.g., Schrauder et al. report n=59 miRNAs with adj. p < 5%, but only the n=25 top hits are shown in Table 2 since data for the remaining 24 are not included in their publication. Please review the criteria carefully (e.g., McDermott do not seem to apply a p-value cutoff, but a rank ordering based on ANN models and Monte Carlo simulations). It should be clearly stated whether only upregulated or up- and down-regulated miRNAs were considered (e.g. using FC>2 or <1/2 in the latter case) and p<5% or p<=5% was used (likewise for FC>2 or >=2; some miRNAs have p=0.05 in Table 2). |
We have provided a detailed explanation of the microRNA data selection criteria for Table 2 in the Results section. |
|
In this context, Table 2 should be updated: For example, no miRNAs are listed for McDermott et al. and only n=3 for Jusoh et al., whereas the published discovery results comprise 76 and 40 miRNAs, respectively. |
We did not include the discovery results of McDermott et al because Supplementary Table 1 contains only a list of altered miRNAs without additional information, and as far as we know, the results in Table 4 pertain to validation rather than discovery. As for Juosh et al, consistent with our approach for the other studies, we included only the miRNAs that showed statistical significance (at least in the not adjusted analysis). |
|
Furthermore, Zavesky et al used two different Ct-cutoffs (Ct<35 and Ct<=40, not Ct>40) and the results of both were merged in Table 2 without indication and explanation how the data was combined. |
We have change from Ct>40 to Ct<=40 in Table 1. Regarding the Ct-cutoffs of Zavesky et al., we decided to include all the miRNAs with a Ct-cutoffs <35, and to minimize data loss, we also added all the miRNAs that had not already been included with a Ct cut-off <=40. |
|
The latter also applies to the nondiscrimination between mature miRNAs originating from the opposite arms of the same precursor miRNA (i.e., nondiscrimination between suffixes “-3p”, “-5p”, “*” or no suffix). This should be mentioned e.g. in a footnote or the text, including how data for a study was merged in this case. |
We have provided an explanation on how we managed with suffixes in the Results section. |
|
|
|
|
Minor points:
|
|
|
According to Table 1, N=6 studies used serum and N=3 blood, but N=7 and 2 are mentioned on page 4. |
We have uniformed to the table 1 the text in the results section about numbers of studies using serum and blood samples. |
|
Percentages on page 5 referring to Figure 3 should be updated (presumably 62.5%, 100% and 50%). |
We have updated the percentage referring to Figure 3 with 62.5%, 100% and 50%. |
|
50 cases/controls are listed for Wu et al. in Table 1, whereas "13 BC patients vs. 10 healthy controls” are described for the discovery part in the publication. |
We have corrected the number of cases and controls in Wu et al. |
|
Godfrey et al.: p-value for 1255a in Table 2 should presumably read p<0.01 instead of <0.001 |
We have replaced <0,001 with <0,01 in Table 2 for MIR 1255a of Godfrey et al. |
|
Please recheck platform names and versions in Table 1 (e.g., Affymetrix GeneChip miRNA 2.0 array in Godfrey et al, Version v3.0, not of 2.1/2.0 of TaqMan Array Human MicroRNA Cards A and B in Zearo et al) |
We rechecked versions of platforms and fixed the problems. |
|
Please replace p>1 with p=1 in Table 2 |
We have replaced p>1 with p=1 in Table 2. |
|
“snRNA” at the end of Table 2 should presumably read U6 “sRNA” |
We have replaced snRNA with U6 snRNA in Table 2. |
|
miRNA 451 in Zavesky et al (Table 1) seems to be from mouse, please indicate this e.g. in a footnote |
We have considered whether to keep mirna 451, given that it is a mouse miRNA. We decided not to compare a mouse miRNA with a human one, so we excluded mirna 451 from Table 2 and the directional graph. |